# Soil Organic Matter Input Promotes Coastal Topsoil Desalinization by Altering the Salt Distribution in the Soil Profile

**Jingsong Li [1,2], Weiliu Li [2,3], Xiaohui Feng [2,3], Xiaojing Liu [2,3], Kai Guo [2,3,*], Fengcui Fan [1], Shengyao Liu [1] and Songnan Jia [1]**

[1] The Institute of Agricultural Information and Economics (IAIE), Hebei Academy of Agriculture and Forestry Sciences, Shijiazhuang 050051, China; lijingsongsjz@163.com (J.L.); njsffc@163.com (F.F.); nkynxs@163.com (S.L.); 18503299303@163.com (S.J.)

[2] Key Laboratory of Agricultural Water Resources, CAS Engineering Laboratory for Efficient Utilization of Saline Resources, Center for Agricultural Resources Research, Institute of Genetics and Developmental Biology, Chinese Academy of Sciences, Shijiazhuang 050021, China; liweiliu1219@163.com (W.L.); fxhcaf@163.com (X.F.); xjliu@sjziam.ac.cn (X.L.)

[3] University of Chinese Academy of Sciences, Beijing 100049, China

* Correspondence: guokai@sjziam.ac.cn

**Abstract:** Organic amendment is an effective method to reclaim salt-affected soil. However, in coastal land with shallow saline groundwater, it is limited known about the mechanism of organic amendment on soil desalinization. Thus, to examine the effect of topsoil organic matter content on soil water/salt transport and distribution, two-year field observations in Bohai coastal land, North China, and soil column experiments simulating salt accumulation and salt leaching were conducted, respectively. There were different organic fertilizer amendment rates in 0–20 cm topsoil, 0% (CK), 50% (OA 0.5), and 100% (OA 1.0) (*w/w*) for soil column experiments. Field observation showed that after organic amendment (OA), the soil's physical structure was improved, and less of the increase in topsoil salt content was observed, with more salt accumulated in deep soil layers during the dry season. In addition, OA greatly promoted salt leaching during the rainy seasons. The results of the soil column tests further indicated that OA treatments significantly inhibited soil evaporation, with less salt accumulated in the topsoil. Although there was no difference in soil water distribution between the CK and OA 0.5 treatment, the topsoil EC for the OA 0.5 treatment was significantly lower than that for CK. During soil water infiltration, the OA 0.5 and OA 1.0 treatments significantly increased the infiltration rates, enhanced the wetting front, and promoted salt leaching to deeper soil layers, compared with CK. The improvement of soil organic amounts could make the soil more self-resistant to the coastal salinization. The findings of this study provide some insights into soil water/salt regulation in heterogeneous soil masses and on the permanent management of coastal saline farmland.

**Keywords:** coastal land; organic amendment; salt leaching soil salinization; soil water/salt transport

## 1. Introduction

Soil salinization causing land degradation has become one of the most serious obstructions for agricultural production [1–3]. There is about 831 million ha of salt-affected land in the world, which accounts for approximately 25% of agricultural land area [4,5]. To fight against the food crisis, it is urgent to utilize salt-affected land resources to overcome the gaps between increasing population and limited farmland resources [6]. Although there have been various approaches to controlling soil salinity in the short term, it is difficult to prevent soil salinization sustainably in both arid inland and coastal areas [7,8].

In coastal land, soil salinization occurs naturally due to seawater intrusion and upward salt accumulation [9]. Along with soil capillary water rise, soluble salts continually

accumulate in surface soil from the salts stored in shallow saline groundwater or deep soil layers [10,11]. Driven by soil evaporation, this water transport determines soil salinization in coastal land. On the contrary, natural rainfall and artificial irrigation result in water infiltration and draining, which contribute to soil salt leaching [12,13]. In coastal agroecosystems, soil evaporation induces salinization and phenological rainfall induces soil desalinization, together dominates the dynamics of soil water and salt distribution [14,15].

Organic matter is an important soil component, which plays an irreplaceable role in soil structure formation, fertilizer retention capacity, and buffering performance, as well as the prevention of soil degradation [16,17]. Previous studies suggested that soil salinity is generally negatively related to soil organic matter content (SOM) [18–20]. Generally, SOM is low in salt-affected land, and poor soil structure and deteriorating soil properties of saline soil have been observed [21,22]. To reclaim salt-affected land, an alternative approach is using exogenous organic amendments to improve SOM [23,24]. It was reported that various organic amendments, such as cattle dung, vermicompost, sludge, leaf manure, crop straw, sugarcane pith, and biochar have shown significant effectiveness in improving soil quality and decreasing soil salinity [25–29]. However, there is still limited known about how the alteration of SOM impacts soil salt content, especially in coastal land.

In coastal land near Bohai bay, North China, there is 1196.3 ha area of land with high potential for agricultural production, while greatly affected by soil salinization [30]. Determined by the semi-humid continental monsoon climate, which is characterized by a short rainy summer and a long dry spring, autumn, and winter, there are basic seasonal dynamics that influence soil salt content [15,31]. In these conditions, there is an obvious dynamic of topsoil salt content, which increases sharply during the dry season because of the high evaporation/precipitation ratio and decreases during the rainy season owing to rainwater infiltration and salt leaching [5,15,32]. To examine the effects of soil organic matter content changes on soil salt distribution, a two-year field experiment on this land was conducted after organic fertilizer input, and the changes of soil properties were measured. In addition, to enhance the regulation mechanism of soil water and salt transport, soil column experiments were conducted to simulate (1) soil evaporation-induced salinization with shallow saline groundwater and (2) freshwater infiltration-induced salt leaching. Soil hydrodynamics characteristics and salt distributions in the soil profile were examined and analyzed. The findings of this study can provide some insights into soil salinization in coastal land and provide effective approaches to reclaiming coastal salt-affected land.

## 2. Materials and Methods

### 2.1. Experimental Site

The study site was located in Bohai coastal salt marsh, Haixing county, Hebei province, North China (117°32′–117°58′ E, 38°19′–38°29′ N), where the groundwater level ranges from 0.8 to 2.0 m with salt content of 10 to 40 g/L [33] (Figure 1). The soil has the texture of silty clay loam, and is classified as an inceptisol, according to the Natural Resources Conservation Service (NRCS) soil classification system. There is a semi-humid continental monsoon climate with an annual average temperature of 15 °C, and annual precipitation of 580 mm, approximately 75% of which occurs in summer, from July to September [31].

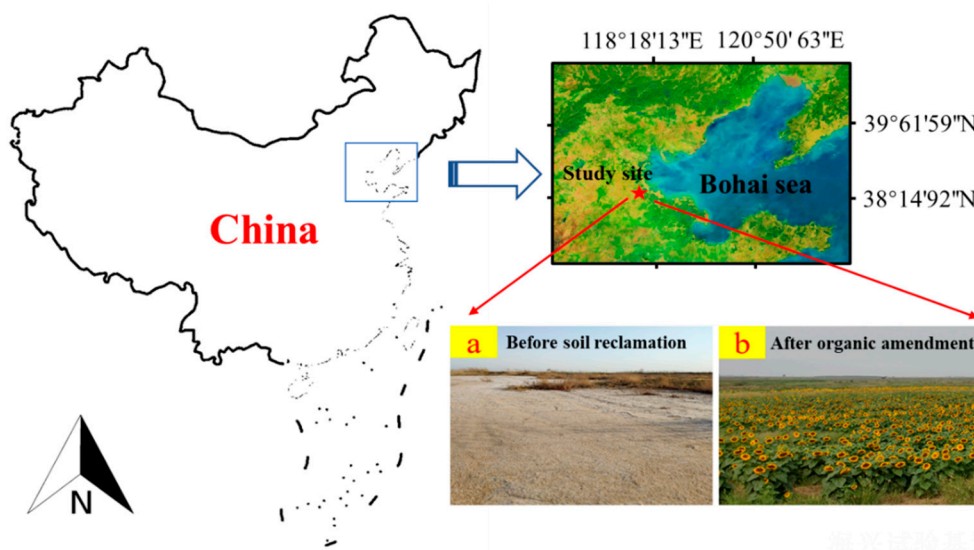

**Figure 1.** The location of the study site and the soil reclamation performance.

*2.2. Field Observation*

The field experiment was carried out from November 2019 to October 2021 at the study site. The material for organic amendment was the "organic fertilizer" product from HuaYu Agricultural Science and Technology CO., LTD. (Handan, China). It was composed of maize and soybean meal material. The main composition for the organic fertilizer included organic matter content $\geq$45%, N + $P_2O$ + $K_2O \geq$ 5%. The organic fertilizer amendment was applied in 3.0 tons (air-dried to 10% water content) per hectare, and after sprinkling, the organic fertilizer was incorporated into 0–0.2 m depth topsoil by a small type of roto-cultivator. After organic fertilizer amendment for 6 months, soil samples were collected to measure the changes in soil properties, and collected continuously each month at intervals of 0.2 m to a depth of 0.6 m with four replicates to measure the soil water and salt profile dynamics.

*2.3. Soil Column Experiments*

Soil column experiments were conducted from October 2019 to June 2020 in the laboratory of Nanpi Eco-Agricultural Experimental Station, Chinese Academy of Sciences. Bulk surface soil samples (0–0.4 m depth) were collected from the study site and the collected soil samples were air-dried (gravimetric water content of 2.9%), ground, and passed through a 2 mm sieve and then stored for experiments. The total salt content of the soil was 1.0% (calculated as grams of water-soluble salt in 100 g of an air-dried soil sample) and the dominant cation and anion in the soil samples were $Na^+$ and $Cl^-$, respectively, together accounting for about 81% of the total salt content. The organic amendment fertilizer was the same as that used in the field experiment (as described in Section 2.2). Before the experiments, the organic fertilizer was oven-dried until the gravimetric water content dropped to 2.9% (the same as that of the soil). The properties of the soil and organic amendment are provided in Table 1.

**Table 1.** Properties of soil, organic fertilizer, and groundwater.

| Properties | Materials | | |
|:---:|:---:|:---:|:---:|
| | Soil Sample | Organic Fertilizer | Groundwater |
| Organic matter (%) | $0.60 \pm 0.02$ | $49.25 \pm 0.45$ | – |
| pH | $6.74 \pm 0.10$ | $5.83 \pm 0.05$ | $7.36 \pm 0.09$ |
| EC (dS /m) | $2.01 \pm 0.03$ | $9.82 \pm 0.64$ | $32.2 \pm 1.37$ |
| Main ion content (g/kg) | | | |
| $HCO_3^-$ | $0.37 \pm 0.09$ | $1.29 \pm 0.28$ | $0.12 \pm 0.02$ |

**Table 1.** *Cont.*

| Properties | Materials | | |
| --- | --- | --- | --- |
| | Soil Sample | Organic Fertilizer | Groundwater |
| $Cl^-$ | $4.96 \pm 0.60$ | $3.90 \pm 0.79$ | $12.27 \pm 1.20$ |
| $SO_4^{2-}$ | $1.25 \pm 0.72$ | $9.59 \pm 1.12$ | $0.18 \pm 0.47$ |
| $Ca^{2+}$ | $0.14 \pm 0.03$ | $1.64 \pm 0.73$ | $0.08 \pm 0.01$ |
| $Mg^{2+}$ | $0.21 \pm 0.09$ | $2.14 \pm 0.54$ | $0.03 \pm 0.01$ |
| $Na^+$ and $K^+$ | $3.40 \pm 0.21$ | $0.53 \pm 0.09$ | $7.95 \pm 0.29$ |

Note: Values represent means $\pm$ S.D.

### 2.3.1. Salt Accumulation Experiment

Soil columns were made of PVC material cylinders that were 100 cm in height and had an internal diameter of 10 cm. At 0–20 cm soil depth on the bottom of each soil column, there were 16 uniform distribution holes (1 mm in diameter) along the wall, in a layout with 4 columns and 4 rows. Filter paper (0.53 mm thickness) was tiled on the inside of each hole and sealed the bottom of the column to prevent soil outflow (Figure 2). A plastic bucket 50 cm in height with a 50 cm diameter was used to load saline water (simulating groundwater). The soil column was vertically placed in the bucket through a 10 cm diameter hole in the cap, and soil was packed in layers at 10 cm intervals, with a homogeneous 1.4 g/cm$^3$ bulk density in all soil layers. The subsoil layer (20–100 cm in height) was entirely composed of coastal saline soil. The topsoil layer (0–20 cm in height) was composed of coastal saline soil with different amounts of organic fertilizer amendment, based on the following organic amendment/soil weight ratios: 0% (CK), 50% (OA 0.5), and 100% (OA 1.0).

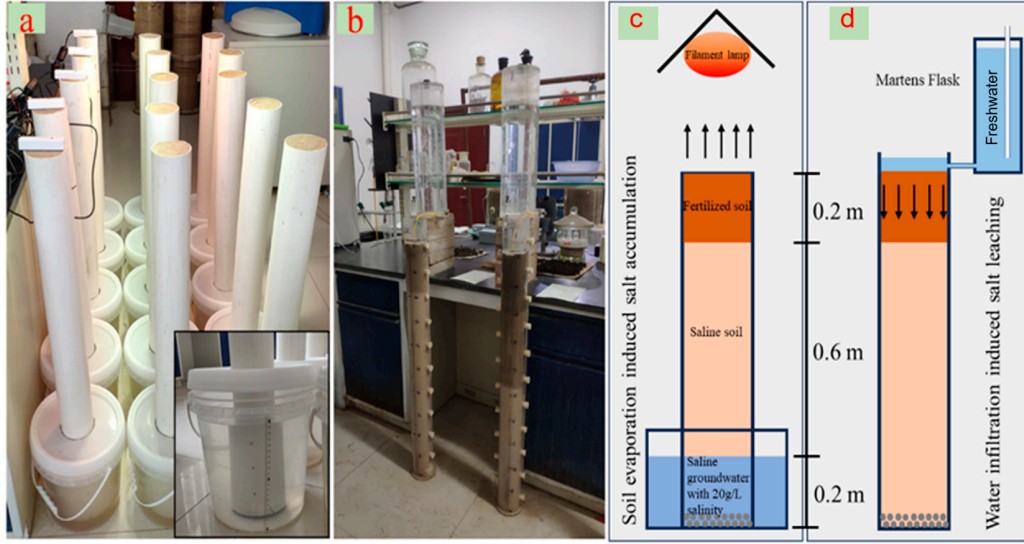

**Figure 2.** Photographs of soil column experiments: (**a**) salt accumulation experiment; (**b**) salt leaching experiment, and their sketches (**c**) salt accumulation experiment; (**d**) salt leaching experiment, respectively.

Soil sensors (TEROS12, METER Group, Inc., Pullman, MA, USA) were inserted into the topsoil for each treatment to examine the dynamics of soil moisture and EC, with an data acquisition unit (EM50, METER Group, Inc., Pullman, MA, USA) collecting the data once an hour. A total of 20 g/L saline water (sea salt dissolved with freshwater, properties as shown in Table 1) was poured into the bucket until a 20 cm height water level was reached, simulating an 80 cm depth groundwater level for the soil column experiment. The saline water gradually intruded into the soil profile through the filter paper at the bottom and the holes of the soil column. Sodium lamp lights with a power of 300 W (40 cm over the surface soil) were used to drive soil evaporation. Throughout the experiment, the

sodium lamp lights were kept on and the average temperature of the surface soil stayed at about 25 °C. At 1, 10, 20, 30, 40, and 50 days after the test began, we replenished the saline water in the bucket until the initial 20 cm height water level was reached, and recorded the volume of the supplied water to calculate the groundwater consumption. At the end of the experiment (50 days in total), 0–20, 20–40, 40–60, 60–80, and 80–100 cm depth soil layer samples were collected with the digging method to measure the soil's gravimetric water content (SWC), electronic conductivity (EC), and main soil cation and anion content. Every treatment was performed in three duplicates.

At the beginning and end of the experiment, we weighed the soil column and calculated the D-value to determine the weight of the solutes that were stored in the soil profile.

Soil evaporation (ET) was calculated using the following equation:

$$ET = (Q - Wi/r)/S \tag{1}$$

where Q indicates the total groundwater consumption in the experiment, *Wi* indicates the weight of the solute that was stored in the soil profile, *r* indicates the ratio of solute weight, and *S* indicates the bottom area of the soil column.

### 2.3.2. Salt Leaching Experiment

The soil columns were 100 cm in height and had an internal diameter of 10 cm (Figure 2). The bottoms of the cylinders were sealed with a circular Plexiglass sheet (with a diameter of 13 cm). There were 12 holes (with a diameter of 0.2 mm) drilled at the bottoms of the cylinders for drainage. In addition, holes (with a diameter of 1.5 cm) were drilled along the walls of the cylinders at 5 cm intervals. During the salt leaching experiment, holes were blocked with rubber stoppers to prevent the leakage of water (as shown in Figure 2b). Additional details on the water infiltration equipment are presented in Guo et al., 2019. The columns were loaded with soil samples at a 1.4 g/cm$^3$ bulk density for all soil layers. Treatments were the same as those in the salt accumulation experiment. A piece of filter paper (0.53 mm thickness) was tiled on the surface of the soil for each soil column. During the experiment, a 2 cm high water level above the surface soil was produced with a Mariotte flask, in which freshwater was used to simulate rainwater. The data on the wetting front and infiltration capacity were collected through the scale plate, respectively, in the soil column and Mariotte flask. We recorded the data 10 times in each interval of 2, 5, 10, 30, and 60 min and 20 times in intervals of 120 min until the test ended (3000 min in total). Throughout the experiment, the average temperature for the laboratory was about 25 °C. After the experiment, soil samples were collected to measure the soil gravimetric water content (SWC), electronic conductivity (EC), and main soil cation and anion content using an interval of 10 cm in depth. Each treatment was performed in three duplicates.

### 2.4. Measurement

After organic amendment in the field, soil samples were collected at intervals of 0.2 m to 0.6 m in depth. Steel rings (100 cm$^3$) were used to measure soil bulk density (BD), saturated hydraulic conductivity (SHC), and porosity indicators. An undisturbed soil block was used for soil aggregate analysis; disturbed soil was used to determine the soil organic carbon content (SOC). In the laboratory, soil cores were saturated by capillarity for 24 h and then weighed and subjected to matric suction of 30 and 100 KPa using a tension table, and their volumetric water content was determined. Finally, the cores were oven-dried at 105 °C for 24 h to measure soil bulk density (BD). Soil total porosity (TP) was calculated with Equation (2), assuming a soil particle density (ds) of 2.65 g/cm$^3$ [34].

$$TP = (1-BD/ds) * 100 \tag{2}$$

Soil water storage capacity (SWSC) was calculated with Equation (3), where $\theta_{FC}$ is the volumetric water content at full capacity, which is considered equal to $\theta_{30KPa}$ [35].

$$SWSC = \theta_{FC}/TP \tag{3}$$

Soil aggregate composition was determined using the standard wetting method described by Kemper and Chepil [36]. Soil samples were air-dried and subjected to a vertical oscillator apparatus that was equipped with a set of seven sieves with the following opening sizes: 2.00, 1.00, 0.50, 0.25, 0.11, and 0.05 mm. The water-stable aggregate percentage in different size classes was calculated using the relationship of the distributed aggregate mass on the sieves and the total mass of the soil samples. Soil mean weight diameter (MWD) was calculated using Equation (4):

$$MWD = \sum X_i * W_i \tag{4}$$

where $X_i$ indicates the mean diameter of each size fraction, and $W_i$ indicates the proportion of soil aggregate weights in the corresponding size.

For both field and indoor experiments, the gravimetric soil water content (WC) was measured by weighing the fresh soil sample before and after oven-drying at 105 °C. Soil electrical conductivity (EC) was analyzed in 1:5 soil water extracts with a conductivity meter (B-173, Horiba. Ltd., kyoto, Japan), as described in Guo and Liu [37]. The main soil cation (sum of $Na^+$ and $K^+$, $Ca^{2+}$, $Mg^{2+}$) and anion ($Cl^-$, $HCO^-$, $SO_4^{2-}$) contents were determined by titration by passing dry soil through the 1 mm sieve [38]. The soil organic matter content was measured with the method of titration with potassium dichromate described in Dong [39]. The values of topsoil field capacity were measured with different soil samples artificially loaded in ring knives (5 cm in diameter, 100 $cm^3$ in volume) using the method of Cassel [40].

The data on air temperature and precipitation were obtained from a weather station (INSENTEK, Beijing, China) at the study site (Figure 3).

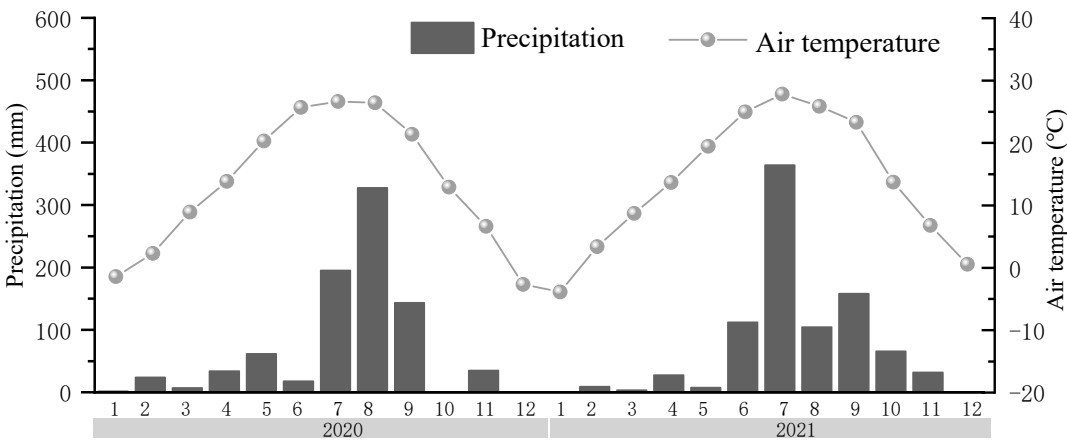

**Figure 3.** The precipitation and air temperature in each month at the study site.

### 2.5. Data Analysis

One-way ANOVA was performed to test the differences between the different treatments. The term significant indicates differences for which $p \leq 0.05$. The statistical procedures were performed using SPSS 16.0 software (SPSS Inc., Chicago, IL, USA).

## 3. Results

### 3.1. Changes in Soil Properties in Field Experiment

The soil BD of the OA treatment was 1.24 g/$cm^3$ in the 0–20 cm soil layer, significantly lower than that of CK (1.40 g/$cm^3$) (Table 2). Meanwhile, OA significantly increased TP and MA, while it had little influence on ME and MI. The value of SWSC of OA was 0.67, significantly lower than that of CK (0.73). In addition, OA significantly increased the soil SHC from 1.84 cm/h (CK) to 5.43 cm/h.

**Table 2.** Soil bulk density (BD), porosity indicators, total porosity (TP), soil water storage capacity (SWSC), and saturated hydraulic conductivity (SHC).

| Soil Depth (cm) | Treatments | BD (g/cm³) | Porosity Indicator (m³/m³) | | | TP (%) | SWSC (-) | SHC (cm/h) |
| | | | Macro-(MA) | Meso-(ME) | Micro-(MI) | | | |
|---|---|---|---|---|---|---|---|---|
| 0–20 | CK | 1.40 ± 0.02 a | 0.12 ± 0.01 b | 0.05 ± 0.01 a | 0.29 ± 0.02 a | 47.16 ± 1.71 b | 0.73 ± 0.04 a | 1.84 ± 0.34 b |
| 0–20 | OA | 1.24 ± 0.03 b | 0.17 ± 0.01 a | 0.07 ± 0.01 a | 0.28 ± 0.01 a | 53.21 ± 0.95 a | 0.67 ± 0.02 b | 5.43 ± 1.31 a |
| 20–40 | CK/OA | 1.46 ± 0.06 | 0.06 ± 0.02 | 0.05 ± 0.01 | 0.34 ± 0.09 | 44.90 ± 2.03 | 0.89 ± 0.05 | 1.75 ± 0.14 |
| 40–60 | CK/OA | 1.49 ± 0.02 | 0.09 ± 0.03 | 0.04 ± 0.02 | 0.31 ± 0.07 | 43.77 ± 1.16 | 0.92 ± 0.02 | 1.69 ± 0.47 |

Note: Values represent means ± SE (n = 4). Different letters within columns indicate significant differences for treatments ($p < 0.05$).

OA treatment significantly enhanced SOC from 5.49 g/kg (CK) to 6.69 g/kg, thereby increasing LA + SA and reducing MI and CS (Table 3). Furthermore, the MWD of OA was 37.55 mm, significantly higher than that of CK (24.65 mm). For soil layers of 20–40 and 40–60 cm, there was no difference in the soil properties between OA and CK.

**Table 3.** Soil organic carbon (SOC), soil aggregate size percentages, and mean weight diameter (WMD) under different straw treatments.

| Soil Depth (cm) | Treatments | SOC (g/kg) | Soil Aggregate Size Percentage (%) | | | | | MWD (mm) |
| | | | LA | SA | LA + SA | MI | CS | |
|---|---|---|---|---|---|---|---|---|
| 0–20 | CK | 5.49 ± 0.41 b | 4.30 ± 1.53 b | 31.46 ± 1.80 b | 35.76 ± 1.72 b | 30.22 ± 1.75 a | 33.52 ± 3.97 a | 24.65 ± 2.12 b |
| 0–20 | OA | 6.69 ± 0.23 a | 7.82 ± 1.60 a | 49.09 ± 3.18 a | 56.91 ± 3.93 a | 21.40 ± 1.50 b | 21.69 ± 4.88 b | 37.55 ± 3.23 a |
| 20–40 | CK/OA | 4.01 ± 0.62 | 1.68 ± 0.98 | 18.41 ± 2.32 | 20.09 ± 2.44 | 30.14 ± 2.01 | 49.78 ± 6.09 | 19.83 ± 2.42 |
| 40–60 | CK/OA | 3.64 ± 0.21 | 0.78 ± 0.43 | 40.65 ± 8.04 | 4.145 ± 2.57 | 35.97 ± 7.04 | 22.59 ± 3.40 | 21.53 ± 3.05 |

Note: LA, >2.0 mm; SA, 0.25–2.0 mm; LA + SA, >0.25 mm; MI, 0.053–0.25 mm; CS, <0.053 mm. Values represent means ± SE (n = 4). Different letters within columns indicate significant differences for treatments ($p < 0.05$).

### 3.2. Soil Water and Salt Dynamics in Field Experiment

OA treatment did not change the soil water content (SWC) or its dynamics in different soil layers, while it greatly changed the soil salt profile (Figures 4 and 5). With SWC being reduced in the dry seasons, soil salt content (SAC) gradually increased. It was indicated that in the 0–20 cm soil layer, the SAC of CK increased from 3.44 g/kg in November 2019 to 5.84 g/kg in June 2020 and increased from 2.21 g/kg in September 2020 to 6.13 g/kg in June 2021. However, the SAC increase for OA was obviously lower than that for CK. It increased by 1.27 g/kg from November 2019 to June 2020, and increased by 3.11 g/kg from September 2019 to June 2021. On the contrary, higher SAC values in the 20–40 and 40–60 cm soil layers were observed under OA than under CK during the dry seasons, which indicated that compared with CK, more soil salt was accumulated in the deep layers under OA treatment. During the rainy seasons, the SAC under OA treatment in the 0–20 cm layer was obviously lower than that under CK.

### 3.3. Soil Column Experiment: Salt Accumulation

As shown in Figure 6, no volumetric water content or EC changes in the topsoil were observed for the OA 1.0 treatment throughout the experiment. For the CK treatment, the topsoil volumetric water content sharply increased from 3% (the initial value) up to 45% (the stable value) from day 13; meanwhile, the soil EC gradually increased and then stabilized at about 14 dS/m. For the OA 0.5 treatment, the moments of soil moisture and EC changes were both about 5 days later than those for CK. The increase in the speed of volumetric water content and EC changes for the OA 0.5 treatment was also slower than that for CK. The final soil EC in the OA 0.5 treatment was 10 dS/m, much lower than that in the CK treatment.

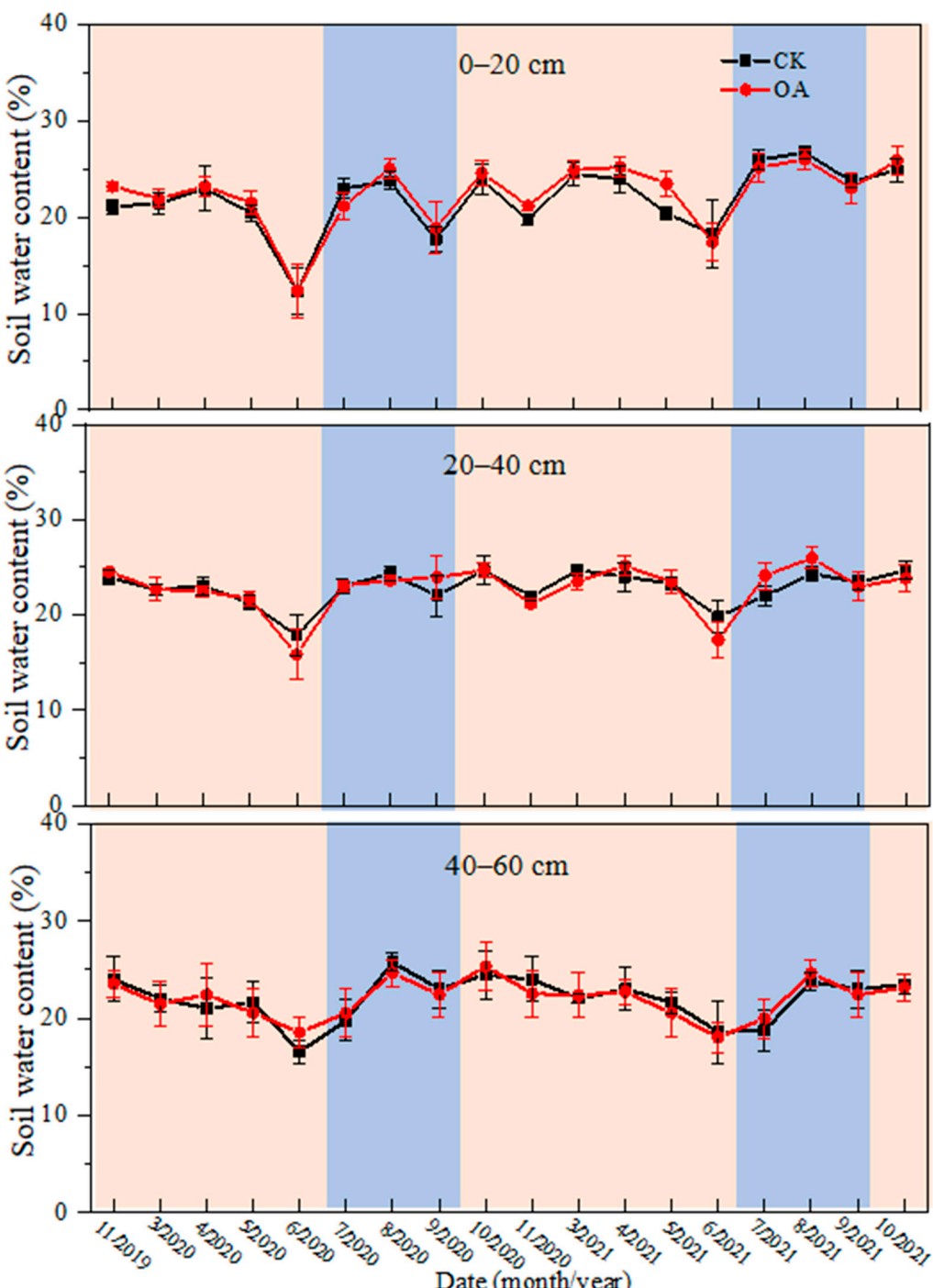

**Figure 4.** The dynamic of soil water content changes in different soil layers for organic amendment in the field experiment. Note: Values represent means ± SE (n = 4); brown color indicates the climate in the dry season and blue color indicates the rainy season.

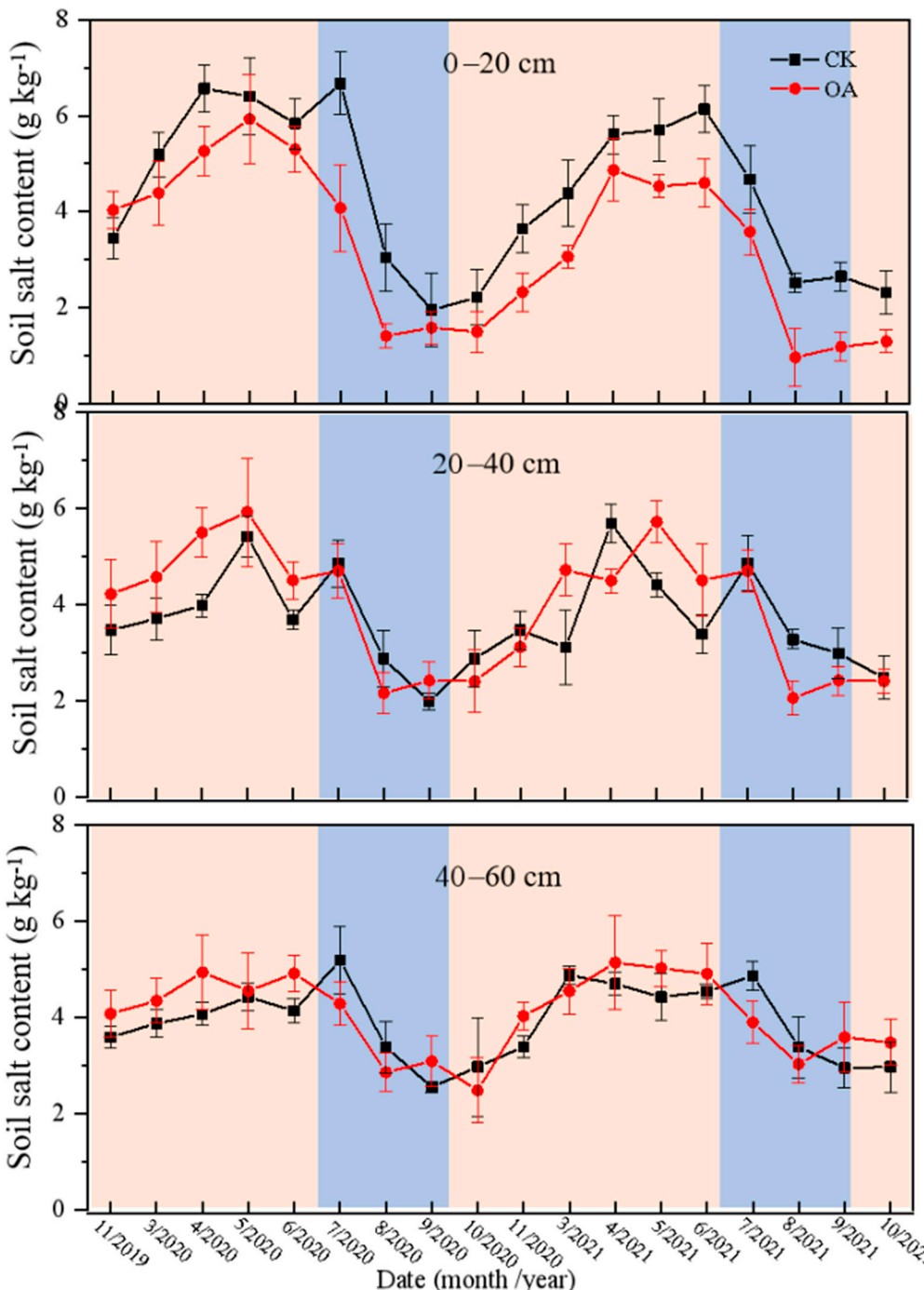

**Figure 5.** The dynamic of soil volumetric salt content changes in different soil layers for organic amendment in the field experiment. Note: Values represent means ± SE (n = 4); brown color indicates the climate in the dry season and blue color indicates the rainy season.

After the salt accumulation experiment, the total value of soil evaporation for CK was 230 mm, significantly higher than that for OA 0.5 (85 mm) and OA 1.0 (50 mm) (Figure 7). There was no significant difference in the soil gravimetric water content and EC between different treatments in the 40–100 cm soil layers (Figure 7). At 0–10 cm, the gravimetric water content of CK (26.7%) and OA 0.5 (25.5%) was significantly higher than that of OA 1.0 (8.7%). Although the gravimetric water content of OA 0.5 was similar to that of CK at 0–20 cm, the topsoil EC of OA 0.5 (9.9 dS/m at 0–10 cm; 5.6 dS/m at 10–20 cm) was significantly lower than that of CK (13.4 dS/m at 0–10 cm; 9.8 dS/m at 10–20 cm).

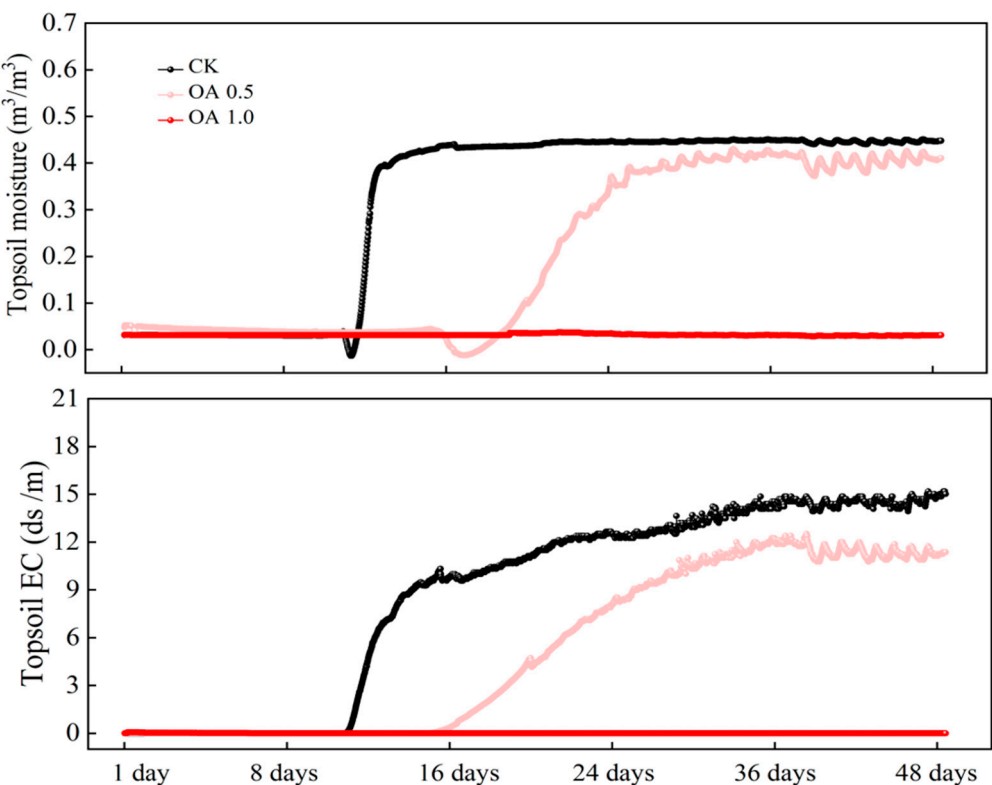

**Figure 6.** The dynamic of soil volumetric water content and EC in topsoil for different treatments in the salt accumulation experiment.

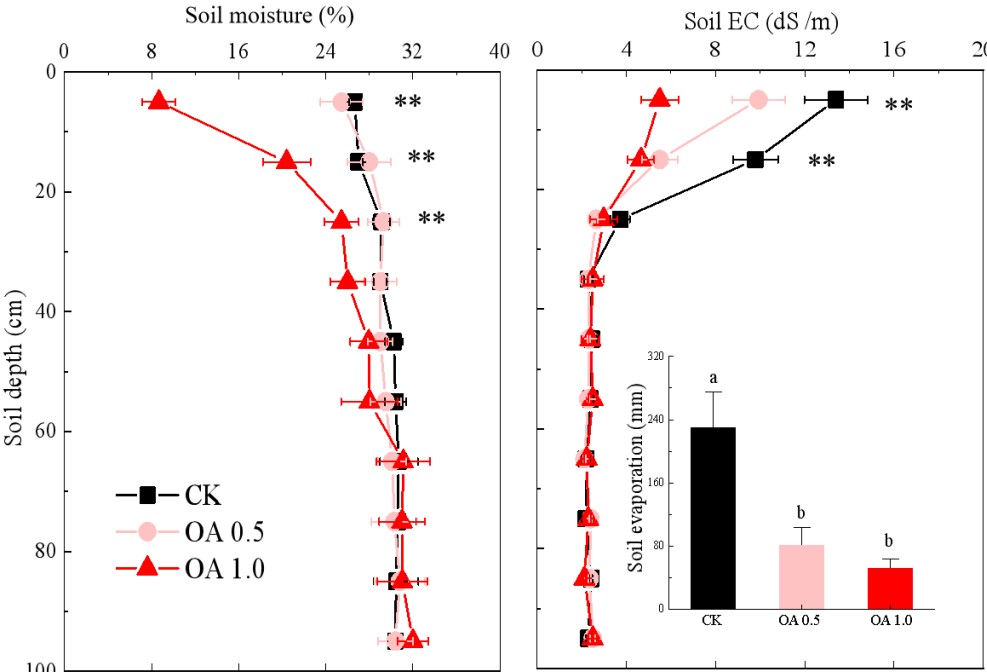

**Figure 7.** Soil water content and EC profile and soil evaporation for different treatments in salt accumulation experiment. Note: **, significance level at $p \leq 0.05$. The bars without the same letter indicate the difference is significant ($p \leq 0.05$). The error bars indicate the standard deviations of the means.

### 3.4. Soil Column Experiment: Salt Leaching

At the beginning of water infiltration, there was no difference in terms of the wetting front among the different treatments, while after a period of time, the wetting front for OA 0.5 and OA 1.0 were obviously higher than that for CK. However, the OA treatments significantly increased the initial infiltration rate from 2.95 mm/min (CK) to 3.01 mm/min (OA 1.0) and 3.58 mm/min (OA 0.5). In addition, the stable infiltration rates OA 1.0 (0.31 mm/min) > OA 0.5 (0.21 mm/min) > CK (0.08 mm/min) (Table 4). At the end of infiltration, the wetting fronts of CK, OA 0.5, and OA 1.0 were 21.7, 52.0, and 48.1 cm, respectively (Figure 8).

**Table 4.** Infiltration rates in initial and stable stages for different treatments in salt leaching experiment.

| Infiltration Rate (mm/min) | CK | OA 0.5 | OA 1.0 |
|---|---|---|---|
| Initial stage | 2.95 ±0.26 c | 3.58 ± 0.44 a | 3.01 ± 0.21 b |
| Stable stage | 0.08 ±0.01 c | 0.21 ± 0.05 b | 0.31 ± 0.03 a |

Note: Values represent means ± S.D. Values followed by different letters are significantly different according to one-way ANOVA ($p \leq 0.05$).

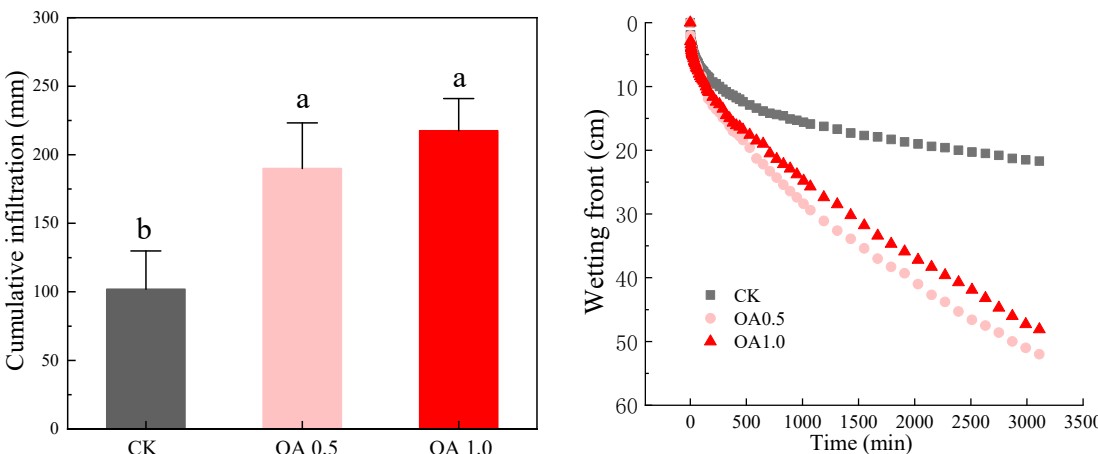

**Figure 8.** Cumulative infiltration and wetting fronts for different treatments in salt leaching experiment. Note: The bars without the same letter indicate the difference is significant ($p \leq 0.05$). The error bars indicate the standard deviations of the means.

After the salt leaching experiment, at depths above the wetting front, the soil gravimetric water content decreased with the increase in soil depth for all treatments (Figure 9). For the topsoil layer, it was found that the soil gravimetric water content ranked as CK (34.0%) < OA 0.5 (43.2%) < OA 1.0 (51.0%). In addition, the soil EC was unchanged at soil layers below the wetting front in the soil profiles for all treatments. The soil EC values at the 0–10 and 10–20 cm depth soil layers were lower than those in the deeper soil layers for all treatments after the salt leaching experiment. For CK, the soil layer with the highest soil EC value was at 20–30 cm, while for OA 0.5 and OA 1.0, the soil salts were accumulated in the 30–40 and 40–50 cm soil layers, respectively.

### 3.5. Changes in Salt Accumulation and Leaching in Topsoil

Before the experiment, OA treatments increased the soil $Ca^{2+}$, $Mg^{2+}$, $SO_4^{2-}$, and $HCO_3^-$ content, while they decreased the total content of $Na^+$ and $K^+$, and the $Cl^-$ content (Table 5). For the CK and OA 0.5 treatments, all cation and anion contents increased after the salt accumulation experiment (EXP1). The soil $Cl^-$ content increased by 128.0% for CK and 52.1% for OA 0.5 and the changes in the percentages of other ion contents in OA 0.5 were also less than those in CK. For the OA 1.0 treatment, the salt accumulation experiment had little influence on the soil cation and anion content. After the salt leaching experiment

(EXP2), the main cation and anion content decreased for all treatments. It changed from 4.96‰ to 4.05‰ without significance regarding the $Cl^-$ content in CK, but the $Cl^-$ content in the OA 0.5 (from 4.26‰ to 2.55‰) and OA 1.0 treatment (from 4.12‰ to 2.91‰) was significantly decreased. The decreases in the sum of $Na^+$ and $K^+$ were 0.72‰ in CK, 0.84‰ in OA 0.5, and 1.10‰ in OA 1.0 after the salt leaching experiment, respectively. Briefly, in coastal land with shallow saline groundwater, there was a negative relationship between soil salt accumulation with SOM in the topsoil and the active impact of salt leaching with SOM (Figure 10).

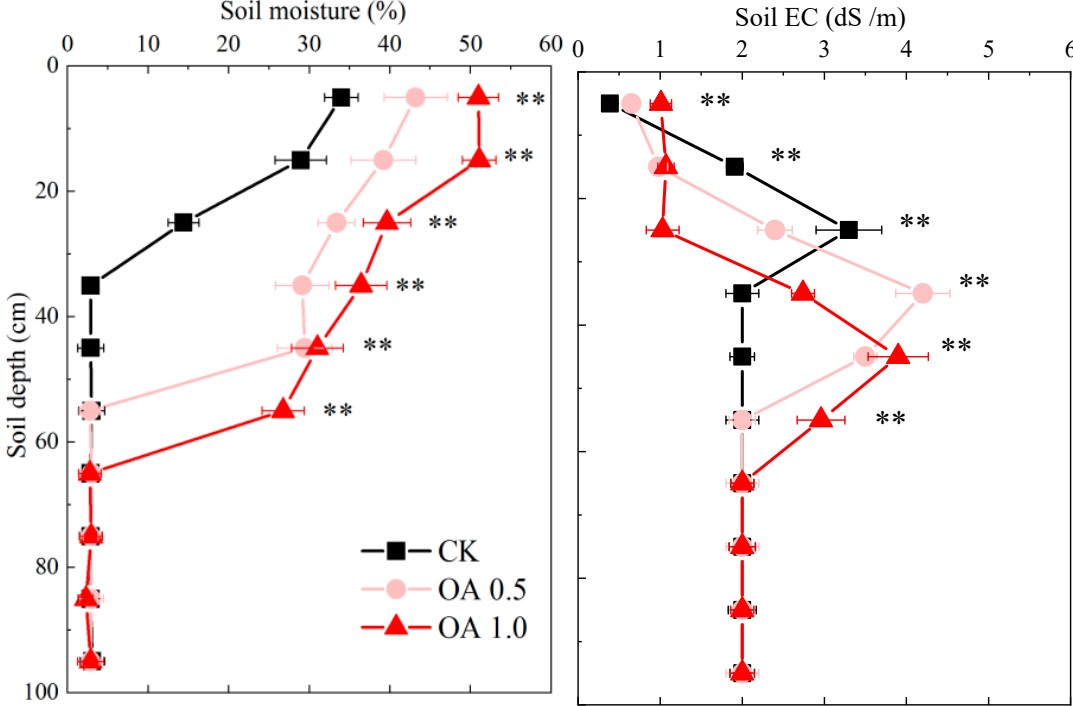

**Figure 9.** Soil water content and EC profile for different treatments in salt leaching experiment. Note: **, significance level at $p \leq 0.05$. The bars without the same letter indicate the difference is significant ($p \leq 0.05$). The error bars indicate the standard deviations of the means.

**Table 5.** Main cation and anion contents of topsoil (0–20 cm) for different treatments before and after salt accumulation (EXP 1) and salt leaching experiments (EXP 2).

| Treatment | Stage | Cation (g kg$^{-1}$) | | | Anion (g kg$^{-1}$) | | |
|---|---|---|---|---|---|---|---|
| | | Na$^+$ and K$^+$ | Ca$^{2+}$ | Mg$^{2+}$ | Cl$^-$ | SO$_4$$^{2-}$ | HCO$_3$$^-$ |
| CK | Before EXP | 3.40 ± 0.21 b | 0.14 ± 0.03 b | 0.21 ± 0.09 b | 4.96 ± 0.60 b | 1.25 ± 0.72 b | 0.37 ± 0.09 ab |
| | After EXP 1 | 7.43 ± 0.24 a | 0.20 ± 0.01 a | 0.48 ± 0.08 a | 11.31 ± 0.79 a | 1.82 ± 0.29 a | 0.42 ± 0.03 a |
| | After EXP 2 | 2.68 ± 0.60 b | 0.04 ± 0.00 c | 0.01 ± 0.00 c | 4.05 ± 0.51 b | 0.14 ± 0.02 c | 0.32 ± 0.02 b |
| OA 0.5 | Before EXP | 2.74 ± 0.23 b | 0.80 ± 0.09 a | 0.73 ± 0.11 a | 4.26 ± 0.61 b | 4.80 ± 0.35 a | 0.55 ± 0.11 ab |
| | After EXP 1 | 4.12 ± 0.67 a | 1.09 ± 0.34 a | 0.84 ± 0.14 a | 6.48 ± 1.05 a | 4.86 ± 0.48 a | 0.67 ± 0.13 a |
| | After EXP 2 | 1.90 ± 0.39 c | 0.06 ± 0.00 b | 0.04 ± 0.01 b | 2.55 ± 0.34 c | 0.17 ± 0.01 b | 0.42 ± 0.03 b |
| OA 1.0 | Before EXP | 2.30 ± 0.15 a | 1.08 ± 0.22 a | 1.02 ± 0.16 a | 4.12 ± 0.33 a | 6.63 ± 1.06 a | 0.67 ± 0.20 a |
| | After EXP 1 | 2.44 ± 0.51 ab | 1.18 ± 0.86 a | 1.05 ± 0.12 a | 4.02 ± 1.50 a | 6.27 ± 0.67 a | 0.81 ± 0.23 a |
| | After EXP 2 | 1.20 ± 0.12 b | 0.18 ± 0.01 b | 0.10 ± 0.01 b | 2.91 ± 0.28 b | 0.48 ± 0.05 b | 0.31 ± 0.01 b |

Values represent means ± S.D. Values in each treatment group followed by different letters are significantly different according to one-way ANOVA ($p \leq 0.05$).

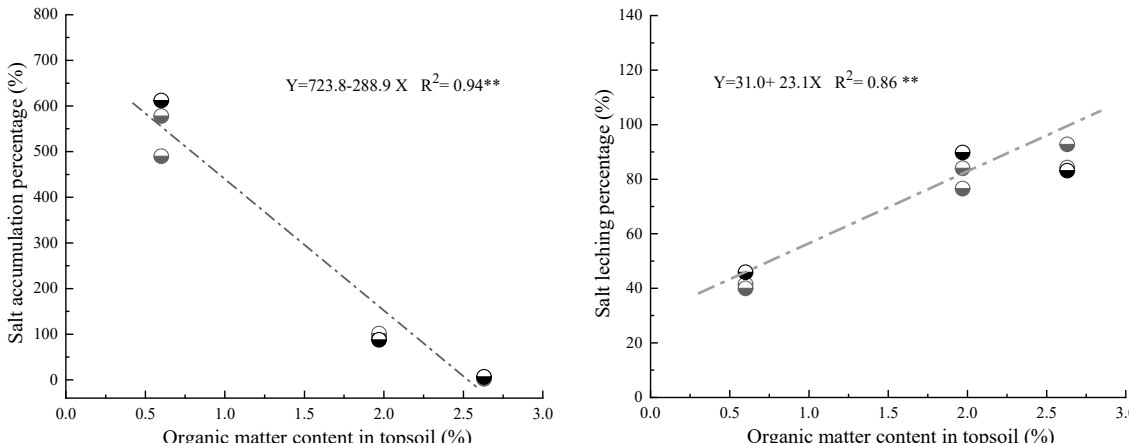

**Figure 10.** The percentages of salt accumulation and leaching for different treatments. Note: **, significance level at $p \leq 0.05$.

## 4. Discussion

In recent years, salt-affected soils have received more and more attention because of the global shortage of arable land and the increasing demand for soil restoration in areas suffering from secondary salinization [41]. In saline land, the high soil salt content in root zones is the main limiting factor for plant growth [42,43]. Moreover, it has been acknowledged that the salt content of topsoil is dynamic and determined by soil water transport [44–47]. In coastal regions, soil salinization mainly results from shallow saline groundwater. The groundwater is like a stockroom which continuously exports the soluble salts into soil profiles. In addition, soil salt is finally accumulated and stored in topsoil during the process of soil evaporation and capillary water rise [46,48].

In the study site, the soil salinity of perennial bare land was extremely high and almost no plants could survive in this land. After a long period of artificial tillage and fertilization, local farmers successfully transformed some of the barren land into cultivated land and were able to plant crops, such as oil sunflower, winter wheat, and corn (Figure 1). It is suggested that compared with the adjacent bare soil, the soil from cultivated land had significantly higher soil organic matter content and lower soil EC. It should be noted that there was no freshwater source for irrigation in this region and the external environmental factors for cultivated land were almost the same as those for bare soil. Therefore, we hypothesized that soil organic matter content played an important role in regulating soil salinity and caused the soil's spatial heterogeneity, as there is a local agricultural saying that "manure battles with soil salinity".

The long-term field observation confirmed that the SOM played an important role in coastal soil salt distribution (Figure 5). In addition, we took the changes in topsoil organic matter content into account to estimate its impacts on soil hydrodynamics and salt distribution in the soil column experiments. As capillary water rising capacity is related to soil porosity, texture, and structure [48], the changes in topsoil structure induced by organic amendment influence soil evaporation as well as salt accumulation. For example, capillary water rise was prevented in the OA 1.0 treatment and there was no salinization observed in the salt accumulation experiment (Figure 6), which indicates that with the increase in SOM, it became hard for stored salts to move upward to the soil surface. Moreover, the lower speed of the soil EC increase and less salt accumulation in the OA 0.5 treatment (Figure 6) were mainly because there was less soil evaporation (Figure 7) compared with that in CK. The results of this study support the view that moderate organic application can inhibit soil water evaporation [48]. In the field, it was also observed that organic amendment reduced soil evaporation, which was partly because this material changed the soil's capacity for heat conduction [49–52].

After water infiltration and draining, salt stored in the topsoil was scoured into the deep soil layer. The study by Mandana et al. [53] indicated that altering the pore system by the addition of organic and inorganic amendments may improve salt leaching as a reclamation strategy. The study by Zhang et al. [54] also demonstrated that biochar and gypsum amendments greatly increased the soil's saturated water content and field water capacity, and the co-application of gypsum and biochar improved saline–alkali soil hydraulic conductivity over a short period. This study indicated that increasing the organic content in the topsoil significantly enhanced the infiltration rate (Table 4), cumulative infiltration, and wetting front (Figure 8). Along with water drainage, the decrease in topsoil EC was greater than that in the deep soil layers, and most salts were depleted from the topsoil and accumulated in the soil near the wetting front (Figure 8). It should be noted that there is unequal infiltration for different parcels of land during precipitation [55,56]. In coastal plains, the rainwater runoff in lands with spatial variability in soil hydraulic properties results in more infiltration occurring in soil with a higher infiltration speed. Thus, the same time as that of freshwater infiltration in this study was used for an accurate simulation of precipitation-induced rainwater infiltration.

As reported in the study by Malak et al. [57], numerous amendments have been employed to mitigate the effects of soil salinization. These amendments are all aimed at strengthening the soil's hydraulic characteristics and promoting the formation of soil aggregates which play a critical role in soil water/salt transport [58]. Nevertheless, organic amendments present the most potential for improving saline soils, both for soil structure improvement and salt leaching. Although in this study there was a difference between the field observation and column experiments, the soil structure improvement was not presented in the indoor experiment, and the increase in SOM in topsoil also significantly changed the salt profile, which suggested that there was a separate regulatory mechanism for soil desalinization by SOM changes and physical structure improvement. With the increasing ability to produce organic matter and the shortage of water resources in agriculture, the establishment of a heterogeneous fertile and desalinized topsoil layer in coastal farmland has been recognized as a new approach to restore coastal salt-affected farmland. And it should be noted that regardless of the organic material types, crop straw, manure, compost, or other organic fertilizer application could result in the similar effects on coastal soil quality improvement and topsoil desalinization through the mechanism proposed in this research.

In summary, this study provides new insight into the mechanism of organic amendment for reclaiming soil in coastal agroecosystems. Soil resistance to salinization was positively associated with the organic matter content in topsoil (Figure 10). Organic amendment can alter soil water transport and decrease the salt content in topsoil during the hydrological processes of soil evaporation and precipitation.

## 5. Conclusions

Organic amendment is an effective and sustainable method for reclaiming coastal saline soil. It promoted the decrease in topsoil salinity mainly through its impacts on water transport and salt distribution in soil profiles. With the increase in topsoil organic matter content, soil salinization was inhibited due to there being less soil evaporation and less salt accumulation from the saline groundwater. Furthermore, organic amendment enhanced the water infiltration and drainage, promoting the leaching of salt out of the topsoil and resulted in more salts accumulated in deep soil layers. We suggest that soil resistance to salinization in coastal regions is largely associated with topsoil organic matter content. This study supports that artificial cultivation of topsoil with high organic matter content is feasible for reducing soil salinity, and it is recommended for the sustainable development of agriculture in coastal salt-affected areas.

**Author Contributions:** Conceptualization, J.L. and K.G.; methodology, W.L.; software, J.L.; validation, X.F., X.L. and F.F.; formal analysis, J.L. and W.L.; investigation, X.F.; resources, K.G.; data curation, J.L.; writing—original draft preparation, J.L. and X.L.; writing—review and editing, K.G., S.L. and S.J.; visualization, X.F.; supervision, X.L.; project administration, K.G. and F.F.; funding acquisition, K.G. and J.L. All authors have read and agreed to the published version of the manuscript.

**Funding:** This research was funded by the National Key Research and Development Program of China (Nos. 2022YFD1900103, 2021YFD1900904); the CAS Engineering Laboratory for Efficient Utilization of Saline Alkali Land Resources, Chinese Academy of Sciences (No. KFJ-PTXM-017); the Talents construction project of science and technology innovation, Hebei Academy of Agriculture and Forestry Sciences (No. C23R1502); and the Modern agricultural industry technology system, Department of Agriculture and Rural Affairs of Hebei Province (Nos. HBCT2023110101, HBCT2023100214).

**Data Availability Statement:** The data are available from the corresponding author on reasonable request.

**Acknowledgments:** Thanks all the workers in Nanpi Eco-Agricultural Experimental Station, and Haixing Saline-alkali Land Resource Efficient Utilization Experimental site, Chinese Academy of Sciences, for the samples collection and measurement.

**Conflicts of Interest:** The authors declare that they have no known competing financial interests or personal relationships that could have appeared to influence the work reported in this paper.

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
