# Peer review of "Soil Organic Matter Input Promotes Coastal Topsoil Desalinization by Altering the Salt Distribution in the Soil Profile"

_agronomy, doi:10.3390/agronomy14050942_

Round 1
Reviewer 1 Report
Comments and Suggestions for Authors
Regarding the ms id agronomy-2977223.
The authors have examined the effect of topsoil organic matter content on soil water/ salt transport and distribution.
The results and discussion are original and relevant for the field of Soil Science.
The results of this ms effectively provide some insights into soil water/salt regulation in heterogeneous soil mass and on the permanent management in coastal saline farmland.
It adds a dataset from a two-year field observation in Bohai coastal land, North China, which can be considered a short-term field experiment. Additionally, the authors have compiled a database from soil column experiments, simulating salt accumulation and leaching. This is highly pertinent and provides a deeper understanding of salt leaching. The experimental design is robust, and there is no need to add further controls. I would suggest incorporating redundancy analysis (RDA) to further enhance the exploration of their main findings.
The conclusions are consistent with the aim of the study regarding the effect of topsoil organic matter content on soil water/salt transport and distribution. All questions were well-addressed in the results and further discussed accordingly. Perhaps the use of redundancy analysis (RDA) could enhance the quality of the results.
I would suggest updating the references. The main theme of this manuscript, comparing organic horizon to soil water/salt mobilization, is a hot topic. Therefore, many studies have been published since 2022 that could be referenced in this manuscript.
Tables and figures follow the authors' guidelines provided in the Agronomy website. All figures have high resolution (300 dpi) and are easy to understand. Tables provide enough information and are well-presented. Even without the use of multivariate analysis, the authors have chosen clear and concise ways to show their main findings.
Author Response
Dear reviewer:
Thanks very much for your comment. (1) We have tried to use the redundancy analysis (RDA) according to the suggestion, but the result was not ideal in this study, and we finally decide not to present it in the manuscript. Overall, thanks for the comment, which is greatly helpful to our further research. (2) New related studies was referenced in the revised manuscript.
Your sincerely,
Correspondence: Kai Guo
Center for Agricultural Resources Research, Institute of Genetics and Developmental Biology, Chinese Academy of Sciences, Shijiazhuang 050021, China
Reviewer 2 Report
Comments and Suggestions for Authors
The authors report a two-year field trial and soil column experiments to understand the impact of soil organic matter application on the salt distribution of the soil profile. This study is relevant to the aims and scope of the journal; however, the current version of the manuscript requires minor revision before it can be approved for publication in this journal.
Comments
- For the abstract, use dry season instead of dry climate (Line 21), remove the conclusion in Lines 27–28, and write the keywords alphabetically (Line 33).
- In Materials and Methods, reduce the font size of the formula to be similar to the text size.
- Why are there no separate values for soil indices 20–40 and 40–60 in Tables 2 and 3?
- The authors should follow the guidelines for writing references.
Comments on the Quality of English Language
· The manuscript needs review for grammatical and typographical errors.
Author Response
Dear reviewer:
Thanks for your comments which is greatly helpful for us and we have revised the manuscript point-by-point.
In table 2 and 3, because the treatment is only for 0-20 cm soil layer, the values of soil indices are the same for both CK and OA. Thus, their values were not separated. For the clearer presentation, we added CK/ OA in the treatment column in tables.
In addition, the quality of English language was improved in new manuscript.
Your sincerely,
Correspondence: Kai Guo
Center for Agricultural Resources Research, Institute of Genetics and Developmental Biology, Chinese Academy of Sciences, Shijiazhuang 050021, China
Reviewer 3 Report
Comments and Suggestions for Authors
Dear colleagues!
You have carried out labor-intensive experimental work worthy of publication in Agronomy. However, there are comments that need to be corrected in the new version of your manuscript. First, we need to provide information about organic fertilizers. Please write in the “Materials and Methods” section what kind of fertilizer it is, its composition (what plant residues, perhaps compost or vermicompost, what mineral additives, perhaps it is provided by some company in China, etc.). I am surprised that your organic fertilizer contains only 9-10% Corg (Table 1). And what is the remaining 90%? Apparently some kind of mineral components. And then it is not an organic, but an organomineral fertilizer. Further, please explain whether the fertilizer was the same in the field and laboratory experiments? Further, the range of doses in a laboratory experiment is absolutely incomprehensible. In the field experiment, you applied 3 t/ha of organic fertilizer and mixed it with the soil in a 20 cm layer. In this case, you obtained BD = 1.2 g/cm3. This means that the mass fraction of the fertilizer in the soil was ONLY 0.125% (0.125*1.2*20=3t/ha)! Why did you use 50% and 100% doses in the laboratory experiment? That is, the doses are 400-800 times HIGHER than in the field? And can your farmers afford such doses (introduce 3*(400-800)=1200-2400 t/ha of fertilizer)!? Further, it is not clear how your magic fertilizer with a carbon content of no more than 10% and a dose of 0.125% of the soil mass could increase the carbon content in the soil from 5.49 to 6.69% i.e. by more than 1%!? (Table 3).In the design of the laboratory experiment, you write about the same filling density BD = 1.4 g/cm3. In the field experiment, on the contrary, the BD 0-20 cm was smaller (1.2 g/cm3). Perhaps this led to a decrease in evaporation and a decrease in salt accumulation. Then why did you compact your fertilizer in soil columns to 1.4 g/cm3? You need to answer these questions in the new version of the manuscript. Apparently you were studying two different mechanisms. In field experiments, this is loosening the soil and reducing evaporation. In the laboratory, this is the mechanism of a capillary barrier, well known and successfully used in soil design (DOI: 10.1134/S1064229321090106). I also ask you to give the units of measurement of indicators ((BD, SHC, etc. in Table 2).) in the new version of the manuscript. I recommend converting the salt content from weight to volume (multiply by DB) in order to also take into account the loosening effect in the Fig. 5. And look through the text, there are typos ("Soil seniors" on page 137, "Soil misture" on Fig. 7, etc.).
April 17, 2024
Best wishes, Your Reviewer
Comments on the Quality of English LanguageThe language is clear and literate, but there are typos.
Author Response
Dear reviewer:
Thanks very much for reviewing our manuscript. We greatly appreciate those comments which is of significance to improve the manuscript. We agree with most of the comments and have revised the manuscript according to the comments. Changes are highlighted in the revised manuscript.
Here is the responds to each point:
1) First, we need to provide information about organic fertilizers. Please write in the “Materials and Methods” section what kind of fertilizer it is, its composition (what plant residues, perhaps compost or vermicompost, what mineral additives, perhaps it is provided by some company in China, etc.)
Respond: Thanks for the comment. We have added the basic information of organic fertilizer in this study, including manufacturer and its composition, in the “Materials and Methods” section in Line 94-97.
2) I am surprised that your organic fertilizer contains only 9-10% Corg (Table 1). And what is the remaining 90%? Apparently some kind of mineral components. And then it is not an organic, but an organomineral fertilizer.
Respond: Thanks very much for your careful checking. There is a number error in Tab.1. We are sorry for this big mistake. Actually, the organic matter for organic fertilizer sample is 49.25%, rather than 9.25%. (How it possible for this value!). We have corrected it in the revised manuscript. For the organic fertilizer, there was 49.25% organic matter and about 5% composition of nutrition N+P2O+K2O, 2% other ions content, as described in Line 97.
3) Please explain whether the fertilizer was the same in the field and laboratory experiments?
Respond: Yes. the fertilizer used in the field is the same with that in laboratory experiments and we have added this description in Line 113.
4) The range of doses in a laboratory experiment is absolutely incomprehensible. In the field experiment, you applied 3 t/ha of organic fertilizer and mixed it with the soil in a 20 cm layer. In this case, you obtained BD = 1.2 g/cm3. This means that the mass fraction of the fertilizer in the soil was ONLY 0.125% (0.125*1.2*20=3t/ha)! Why did you use 50% and 100% doses in the laboratory experiment? That is, the doses are 400-800 times HIGHER than in the field? And can your farmers afford such doses (introduce 3*(400-800)=1200-2400 t/ha of fertilizer)!?
Respond: We agree with this comment and recognize the incomprehensible high input of organic fertilizer in soil column experiments. The aim of indoor examine was to explore the mechanism of organic matter changes on soil salt profile. Actually, we carried out the similar experiments with treatments at 1%, 5%, and 10% (w/w) OA (the proper application for organic fertilizer). While, there was not a significant result. We think it is owing to the lower radiation (simulated by sodium lamp lights) and shorter term than that in the field. Thus, to enlarge its impacts, we have to use this excessive high organic fertilizer amendment treatment, even we knew the value was with none of practical significance, but its rule is of significance for this research.
5) It is not clear how your magic fertilizer with a carbon content of no more than 10% and a dose of 0.125% of the soil mass could increase the carbon content in the soil from 5.49 to 6.69% i.e. by more than 1%!? (Table 3).
Respond: This is an real interesting question. Agreed with the comment, we are also curious about the too large change of SOC in the field experiment. The soil samples were collected after organic amendment for 6 months (as described in Line 101). We think it may be related with the crop (sunflower) cultivation or the activity of soil microbial, which may contribute to the increase of SOC.
6) In the design of the laboratory experiment, you write about the same filling density BD = 1.4 g/cm3. In the field experiment, on the contrary, the BD 0-20 cm was smaller (1.2 g/cm3). Perhaps this led to a decrease in evaporation and a decrease in salt accumulation. Then why did you compact your fertilizer in soil columns to 1.4 g/cm3? You need to answer these questions in the new version of the manuscript. Apparently you were studying two different mechanisms. In field experiments, this is loosening the soil and reducing evaporation. In the laboratory, this is the mechanism of a capillary barrier, well known and successfully used in soil design (DOI: 10.1134/S1064229321090106).
Respond: Thanks for this meaningful comment. We must consider that there is a real difference between the field and indoor experiments. In the field, the soil structural changes by organic amendment, such as soil loosening, played the role in soil water and salt transport. To control the single variable (SOM content) in soil column experiments, we kept the same BD for CK and OA0.5, OA1.0 treatments, which inevitably eliminated the effects of topsoil structure changes. That is because we think the SOM content and its effect on soil structural changes both impacts on soil water /salts profile, separately. If topsoil BD changes, the results maybe similar with that in the previous studies and with little novelty. In addition, although the soil for column experiment was from the field land, it was air-dried and passed 2 mm sieve, which led to the smaller soil particles and higher BD. We think there was not much compaction effects for OA treatments. And the results of salt leaching experiment proved that the high BD in OA treatments did not block the water draining and leaching. It is a pity that we failed to find the reported paper (DOI: 10.1134/S1064229321090106) and compare with this study. If possible, please provide more information, which is likely helpful for the our further discussion.
7) I also ask you to give the units of measurement of indicators ((BD, SHC, etc. in Table 2).) in the new version of the manuscript.
Respond: Thanks for the comment, and we have added the units of soil indicators in revised manuscript.
8) I recommend converting the salt content from weight to volume (multiply by DB) in order to also take into account the loosening effect in the Fig. 5.
Respond: We have converting the salt content from weight to volume in Fig. 5, according to the comment.
9) And look through the text, there are typos ("Soil seniors" on page 137, "Soil misture" on Fig. 7, etc.).
Respond: We have carefully checked the language in manuscript and corrected the spelling errors.
Thanks again for your helpful comments.
Your sincerely.
Correspondence: Kai Guo
Center for Agricultural Resources Research, Institute of Genetics and Developmental Biology, Chinese Academy of Sciences, Shijiazhuang 050021, China
Round 2
Reviewer 3 Report
Comments and Suggestions for Authors
Dear Colleagues!
You have seen all the comments and answered all the questions. From my point of view, the manuscript is now ready for publication. Article DOI: 10.1134/S1064229321090106 is attached.
April 22, 2024.
Best regards, Your Reviewer
